# A Generalist Feeding on Brassicaceae: It Does Not Get Any Better with Selection

**DOI:** 10.3390/plants10050954

**Published:** 2021-05-11

**Authors:** Jacinta M. Zalucki, David G. Heckel, Peng Wang, Suyog Kuwar, Daniel G. Vassão, Lynda Perkins, Myron P. Zalucki

**Affiliations:** 1Centre for Planetary Health and Food Security, Griffith University, Brisbane 4011, Australia; j.zalucki@griffith.edu.au; 2Max Planck Institute for Chemical Ecology, 07745 Jena, Germany; suyogkuwar@gmail.com (S.K.); vassao@ice.mpg.de (D.G.V.); 3School of Biological Sciences, The University of Queensland, Brisbane 4072, Australia; p.wang1@uq.net.au (P.W.); l.perkins@uq.edu.au (L.P.); 4Institute of Bioinformatics and Biotechnology, Savitribai Phule Pune University, Pune 411007, India

**Keywords:** glucosinolates, host specialisation, forced selection, performance assays, pest status

## Abstract

Brassicaceae (Cruciferae) are ostensibly defended in part against generalist insect herbivores by toxic isothiocyanates formed when protoxic glucosinolates are hydrolysed. Based on an analysis of published host records, feeding on Brassicas is widespread by both specialist and generalists in the Lepidoptera. The polyphagous noctuid moth *Helicoverpa armigera* is recorded as a pest on some Brassicas and we attempted to improve performance by artificial selection to, in part, determine if this contributes to pest status. Assays on cabbage and kale versus an artificial diet showed no difference in larval growth rate, development times and pupal weights between the parental and the selected strain after 2, 21 and 29 rounds of selection, nor in behaviour assays after 50 generations. There were large differences between the two Brassicas: performance was better on kale than cabbage, although both were comparable to records for other crop hosts, on which the species is a major pest. We discuss what determines “pest” status.

## 1. Introduction

Plants in the Brassicaceae are often portrayed as being a well-defended plant group, with only specialist insects capable of surviving and developing well, while generalists perform less successfully, e.g., [1]. The front-line defences in Brassicales are glucosinolates (GSL), a group of anionic thioglucosides [2,3]. Although GSL are themselves not toxic, when brought together with activating myrosinases that are maintained in separate compartments in plant cells, for example by the actions of a chewing herbivore, they are hydrolysed to form an array of products. These include prominently the toxic isothiocyanates (ITCs), the main components of the so-called mustard oil bomb [4], as well as other hydrolysis products with potential biological activity, namely epithionitriles, simple nitriles and organic thiocyanates, depending on the reaction cofactors and the GSL side chain [5,6,7]. Due to the reactivities of some of these molecules, especially of ITCs, further intramolecular rearrangements and conjugations to biological nucleophiles are common, leading, for example, to cyclization derivatives, as well as indole carbinols and ascorbigens [5,6,7]. The release of these toxic compounds upon plant tissue damage caused by chewing is considered to be a very effective defence against generalist herbivores [8,9,10], and has been repeatedly shown to adversely affect measures of performance of feeding herbivores, for example using *Arabidopsis thaliana* in assays, e.g., [11,12,13,14].

Of course, various polyphagous Lepidoptera are recorded to feed on plants in the Brassicaceae. Larvae of several lepidopteran generalist herbivores, including *Spodoptera exigua, S. littoralis, Mamestra brassicae, Trichoplusia ni* and *Helicoverpa armigera*, for example, are well known “pests” of Brassica vegetable crops. We take more notice of pest species for obvious reasons. Here, we ask how many species in the Lepidoptera other than “pests” are recorded as feeding on Brassicaceae. Is the handful of generalist pests the exception? For one of the pests, the super generalist *Helicoverpa armigera* [15] larvae have been found to produce and excrete glutathione conjugates of ITCs, suggesting that even this species, presumably lacking biochemical adaptations to feeding on Brassicaceae plants, has some level of GSL detoxification, after the de facto chemical defensive ITCs have been formed during ingestion [16]. 

Although *H. armigera* does not do well on Brassicas, larvae will develop, albeit poorly, on wild type *Arabidopsis,* particularly if allowed to choose feeding sites on whole plants [14]. Cabbage (*B. oleracea* var. *capitata*) is a widely recorded host [15], and larvae of this insect are considered a pest on these crops, particularly in the subcontinent [17,18,19,20,21] and elsewhere [22]. Thakor and Patel [23] readily reared *H. armigera* on cabbage in the laboratory. This suggests that some populations of *H. armigera* do better on Brassica than others, and furthermore, that they might be under selection for better tolerance towards GSL defences, especially where multiple insect generations develop feeding repeatedly on Brassicaceae. That is, it is possible that the pest status of herbivorous species in certain crops may be due to survivors that initially infest a crop and perform better in the next generations, leading to greater pest status. We therefore undertook a laboratory selection experiment to see if the fitness of *H. armigera* on Brassica hosts could be improved, and potentially to help locate the genetic basis for Brassica host use.

Here, we test performance of *H. armigera* on two cultivated Brassicas—common cabbage and kale—that differ markedly in GSL profiles: cabbage with 3-methylsulfinylpropyl (3MSOP)-GSL (absent from kale) and in kale, 2-hydroxy-3-butenyl-(2OH3But) (absent from cabbage) [24]. We undertook selection of the Toowoomba strain (TWB3) of *H. armigera* to see if we could select for “better” performance starting in October 2012 and maintained the trial for 50 generations to November 2019. We designated this “cabbage fed” strain as TKF (Toowoomba “Kohlfütterung”), e.g., refer to 4.3.2.

We undertook three series of rearing assays of both strains: the parental TWB3 and the TKF strains, fed on either artificial diet, cabbage and kale, in January 2013, August 2015 and November 2016, after 2, 21, and 29 rounds of selection, respectively. We assessed several performance measures of herbivorous insects on artificial diet and plant material: larval weight gain, time to complete developmental stages, survival, pupal weight and adult emergence. We undertook simple choice assays to see if feeding preference had changed in first instars (November 2016 and October 2019). We expected TKF to progressively survive and grow better on Brassica plant material and/or show a feeding preference for these plants. Surprisingly, none or only very minor effects were found, but there were major differences in performance measures between the Brassica plants tested that remained relatively consistent in both strains. We discuss the implications of these findings.

## 2. Results

### 2.1. Brassica Host Use 

Globally, some 317 species of Lepidoptera across 20 families are recorded feeding on Brassicaceae (Cruciferae) in the HOSTS database [25] (Table 1). Some families (Pieridae, Pyralidae and Yponomeutidae) have a high number of crucifer feeding specialists (as expected), but overall, only 21% of moths and butterflies that fed on crucifers were specialists and 26% were oligophagous. What is surprising was the high percentage of species that were polyphagous (56%) recorded on Brassicas with the Noctuidae (97 species), Arctiidae (17) and Pyralidae (17) dominating. Of the polyphagous species, 21 of these (13%) are noted as pests, 11 specialists (13%) are noted as pests and 1 oligophagous species (2%) is noted as a pest. 

### 2.2. Survival

Early-stage survival in Petri dish assays was high on diet (90–96%) (Figure 1a) and generally lower on cabbage (57–84%) than kale (65–84%). There was no consistent effect of strain and year (Figure 1a). Ignoring the first assessment, survival to adulthood was generally better on diet (62–80%) than plant (42–76%) but declined over time for diet (Figure 1b). The very poor survival to adulthood on plant material in the first assessment (1–5%) was likely an artefact of rearing animals together; even though containers were large and excess food was available, cannibalism was extensive. Therefore, experiments were conducted utilising individualised rearing in subsequent tests.

### 2.3. Early Larval Weight Gain

Larvae gained the most weight on diet, followed by kale, and grew the slowest on cabbage (Figure 2). Year and food significantly affected the variation in larval weight, but not strain. Larval weight in 2015 was significantly greater than 2013 (P = 0.042427), and larval weight was significantly less on cabbage (P = 0.002044) and kale (P = 0.011775) than on artificial diet. Some 13% of the variation in larval weight left over after fitting fixed effects is due to moth family.

### 2.4. Days to Pupation

All data pertaining to pupae were based on data collected after 21 and 29 rounds of selection. Variation in the time taken to pupate is significantly affected by food but not by year, strain or gender. The number of days to pupation was significantly greater for larvae on kale (P = 0.016323) and on cabbage (P = 0.001992) than on diet. Larvae completed development faster on artificial diet (mean ± SD, 12.0 d ± 1.41), followed by kale (14.3 d ± 1.61) and slowest on cabbage (17.4 ± 2.50 d) (Figure 3a). Some 24% of variation in response leftover after fitting fixed effects is due to moth family. 

### 2.5. Pupal Weight

The variance in pupal weight is significantly affected by food, but not by year, strain or gender. Pupal weight was significantly less for larvae raised on kale (P = 0.003708) and cabbage (P = 0.001939) than on diet. Pupae were heaviest on diet (mean ± SD) (324 ± 33 mg), followed by kale (251 ± 38 mg) and cabbage (217 ± 42 mg) (Figure 3b). Only 6% of variation in response leftover after fitting the fixed effects is due to family.

### 2.6. Time as a Pupa

Variation in time as a pupa (data not shown) is significantly affected by year, food and gender but not by strain. On average, females took a day less to complete the pupal stage (7.8 ± 0.67 d) than males (8.7 ± 0.75 d). Additionally, as with pupal weight, only 6% of variation in response leftover after fitting fixed effects is due to family.

### 2.7. Leaving and Choice Assays

In the leaving assay after 29 rounds of selection, a higher proportion of TWB strain neonates abandoned kale (21% ± 4%, SE, *n* = 6) and cabbage (30% ± 7%) than TKF neonates did—17% (±4%, SE, n = 6) and 15% (±3%), respectively (Appendix A). However, an analysis on the proportion of larvae that had stayed on the plant disc (1-proportion leaving) at 3 h (logit transformed to give the variable a normal distribution: Shapiro–Wilk normality test, W = 0.9634, P = 0.2734) by two-way ANOVA showed that neither the strain nor plant (cabbage or kale) significantly affected whether the larvae stayed or left the plant disc at 3 h. When retesting those that left a second time, the rates of leaving for TWB on kale (50% ± 11%) and cabbage (32% ± 11%) were higher than in the first test, and higher than TKF, with 15% (±7%) and 3% (±3) leaving kale and cabbage, respectively. However, the number of larvae being retested was too small for analysis. After 49 generations, in October 2019, we undertook a feeding choice assay on the parental and selected strain. After 48 h, both strains “preferred” the artificial diet, with comparable proportions found on cabbage and kale for each strain (Table 2). A multinomial model fitted to the data showed the best model was Site Found~Plant, i.e., plant (cabbage or kale) is a significant contributing factor (P = 0.0439), but strain was not. We discontinued the experiment and did not run performance assays, as we had expected the TKF strain to differ from the parental strain.

### 2.8. Comparative Performance on Hosts

Larval performance measures on both cabbage and kale were comparable to data published for other *H. armigera* plant hosts (Table 3), including host crop species on which it is a major pest (e.g., Pigeon pea). There is wide variation in measures of so-called performance but as they are all undertaken using plant parts in the laboratory, we make no attempt to analyse these data here.

## 3. Discussion

Following previous observations that some populations of *H. armigera* perform better than others on the glucosinolate (GSL)-defended plants of the Brassicaceae, we hypothesized that continuous selection might select for better insect tolerance towards GSL defences, with surviving insects and their descendants performing better and leading to greater potential pest status. However, here, we found no effect of selection by exposure of this highly polyphagous pest species to such “sub-optimal” chemically defended hosts on key performance traits across multiple generations. The selected strain (TKF) did not differ from the parental (TWB) strain in survival, early instar growth, time to pupation and time in pupal stage, even after 29 generations of selection. Under laboratory conditions, *H. armigera* survived and grew reasonably well on host plants that produce almost insecticide-like toxins, the pungent GSL-derived isothiocyanates (ITCs) [28]. Larval performance on both cabbage and kale were comparable to data published for other *H. armigera* plant hosts (Table 3), including host crop species on which it is a major pest. The wide variation in measures of so-called performance can be due to many factors [29] but as they are all undertaken using plant parts in the laboratory, they have little bearing on how a population will develop in the field, where natural enemies are present, and when larvae can self-select, to some extent, where and on what they will feed (e.g., [30]). As is the norm, most/all Heliothinae feed on plant reproductive structures [31] and, perhaps not surprisingly, in laboratory assays, larvae grew better on plant reproductive structures (e.g., [32,33,34,35,36]). 

Being a pest is not a function of host specialisation. Feeding on brassicas and being a pest on such crops is just as likely in host specialists as it is in generalists (Table 2). We appreciate data for host use have limitations. The Diamondback moth (DBM), *Plutella xylostella*, is widely acknowledged as a host specialist and key pest of brassicas (e.g., [37]), yet in the HOSTS database, it is recorded on 11 plant families—most would say it is polyphagous on this basis—and 103 plant species, but 84% of these are crucifers, and we retained its designation as a specialist on this criterion. The occurrence of DBM on other plant families may be a spill over given its high numbers in brassica crops and at least one of these represented a localised host expansion on to peas [38,39,40]. Therefore, how does an extreme generalist, *H. armigera*, lacking specific adaptations to brassica hosts, reach pest status on brassica crops?

Pest status is a numbers game and, to a large extent, on the plant part damaged. Generally high pest densities are more problematic but even low numbers of a pest feeding on parts that are to be sold at market may greatly reduce the quality and price. Although brassicas are not preferred oviposition hosts of *H. armigera* in cage choice tests [41,42], if large areas of such non-preferred hosts are available, with little else in the landscape, moths will lay eggs on these plants [43]. Early-stage survival is not high on vegetative plant growth stages, even on plants that *H. armigera* is a key pest, e.g., cotton [44]. Survival improves on flowering plants [34], even if they are genetically modified to express Bt toxins [36,45]. By selecting less toxic plant parts and cannibalism of eggs, survival of early stages can greatly improve [45,46]. If populations in an area are high because of local climate (e.g., [47]) and host plants (e.g., [48,49]), or migration [50], the large number of moths laying eggs on available hosts may increase the rate of cannibalism; first hatching larvae are more likely to eat eggs that are about to hatch [46]. Damage to reproductive structures can greatly diminish the value of the product and, if these structures are available, egg placement and neonate behaviour will take them there [51,52]. Under these conditions, sufficient larvae will survive to older instars. These stages often appear better able to deal with plant toxins [53,54,55] and of course, pest status increases as larvae consume more, particularly in the last instar [56]. With resistance to insecticides in this species [57] and the sample, spray and pray approach to pest management that effectively removes natural enemies [58] and “pest status” is much more likely. It is these processes that probably lead to local pest status on non-preferred and sub-optimal hosts.

In our experiments, we did find significant year-to-year effects (= when the tests were run), but without a clear link to adaptive effects. Although plants were grown under tight protocols, even subtle differences may be significant, such as “time” of year. Assessments were run in January 2013 (after 2 rounds of selection), August 2015 (21 rounds) and November 2016 (29 round) with the final choice test in October 2019 (49 rounds). Could some of the year effects simply reflect differences between plants due to time of year? The only other effect over time was an apparent decline in performance traits (see Figure 1, Figure 2 and Figure 3 in results) which may reflect the nature of inbreeding and maintaining isofemale lines [59]. The effect occurred in both strains.

Although the moth “Family” effect was generally small (ca. 6%), it was higher for time to pupation (24%), suggesting some of the effect of host is affected by genetic traits and that there is standing genetic variation for performance on hosts. Perhaps a more targeted selection of such families might be able to shift host plant use?

Behavioural assays showed that cabbage was not a preferred plant in the leaving or choice assay against artificial diet. Larvae were marginally more likely to leave cabbage than kale and they were more likely to be found on kale than cabbage. These larval behaviours were correlated with performance.

The strongest effects were observed in differences between herbivore development on cabbage and kale, especially in comparison to the pinto bean-based artificial diet routinely used for larval rearing. Performance measures were generally better for kale than cabbage. As in most domesticated plants, the chemical toxicity level of our current crucifer crops is low relative to wild types [60]; however, in this case, the two brassicas have very different chemical profiles [24]. The toxicity of isothiocyanates (ITC) is believed to derive from the reaction of the electrophilic ITC group with the tripeptide glutathione (GSH), resulting in its depletion and instigating other metabolic consequences [61], and with amino acid residues of proteins, leading to cleavage of disulfide bonds and secondary/tertiary structural changes [62]. Schramm et al. [16] showed that conjugation with GSH is a common post-ingestion metabolic pathway for mitigating the toxicity of glucosinolate-derived ITCs in generalist lepidopteran herbivores feeding on *Arabidopsis*, including *H. armigera*. The full GSH-ITC conjugate and its downstream CysGly- and Cys conjugates were the major ITC-derived metabolites produced by the generalist *S. littoralis*. Presumably, this ability to conjugate ITC is more easily saturated or not fully activated in neonates and they can survive only lower doses of ITCs. Although plants have a formidable array of defences, insects do successfully feed on plants, albeit poorly in the first instar [63]. Their eavesdropping on metabolic changes within plants and responding via behaviour and induction of their own detoxification systems, enables them to overcome plant defences from time to time [14,30,64,65]. We investigate the physiological effects and costs of feeding on cabbage and kale in a companion paper [24].

## 4. Materials and Methods

### 4.1. Brassica Feeding Lepidoptera

We searched HOSTS (a Database of the World’s Lepidopteran Hostplants: (https://www.nhm.ac.uk/our-science/data/hostplants/ (accessed on 15 August 2020) [25] for all Brassicaceae (Cruciferae) feeding records). It should be noted that this database uses the old classification, Cruciferae, which is now Brassicaceae [66]. If a lepidopteran was recorded as feeding on a Cruciferae, we downloaded all the host plant records for that species. We take these records at face value. We tallied the number of host families and plant species recorded. We calculated the proportion of families utilised that were Cruciferae and the proportion of host plant species recorded as hosts that were Cruciferae. Each lepidopteran recorded as feeding on Cruciferae was classified as either a specialist (monophagous), generalist (polyphagous) or oligophagous based on the following criteria: the % of host plant families used in the Cruciferae and the % of host plant species used in the Cruciferae:Host family (x) Host species (y) Specialist: 9 < x < 100 56 < y < 100Oligophagous: 13 < x < 50 2 < y < 50Generalist: 1 < x < 33 1 < y < 23

### 4.2. Performance on Different Hosts

Published studies of performance of *H. armigera*, perhaps not surprisingly, report a wide variation in fitness traits or performance measures and tend to be fixated on artificial diets [29]. We took a subset of these studies to compare to our findings. The papers had to report data on larvae reared on plant material (not various artificial diets) and at least include pupal weight.

### 4.3. Assays Assessing the Effects of Selection

#### 4.3.1. Plants

Cabbage (*Brassica oleracea* var capitata, White cabbage “Gloria” (F1 hybrid) from Daehnfeldt Seeds) and kale (*Brassica napus*, Rape broadleaf Essex Salad Sproutin, from B&B World Seeds) were grown in trays (58 × 32 × 11.5 cm) in a peat-based substrate (Klasmann Kultursubstrat TS1, Geeste-Grob Hesepe, Germany) under greenhouse conditions at 21–23 °C, 50–60% RH and 14: 10 L:D photoperiod. Each tray contained approximately 60 plants. Multiple trays were established every 7 d ensuring similar quality food was available for experiments. Plants were used when they were ca. 6 weeks old.

#### 4.3.2. Insects

The *H. armigera* were from the Toowoomba strain (TWB3) maintained on a pinto bean diet at 28–30 °C. The protocol for maintenance of the insects can be found in [67]. We undertook selection of the TWB3 strain to see if we could determine the genetic basis for cabbage feeding and select for “better” performance. Larvae from isofemale lines were exposed to cabbage leaves in Petri dishes (9 cm diameter) and survivors to the third instar were then reared on diet, mated and the offspring of single pair mating re-exposed to cabbage for each generation starting in October 2012 and maintained for 50 generations until November 2019. We designated this putative “cabbage feeding” strain as TKF (Toowoomba Kohlfütterung).

For each performance assessment experiment, we set up single pair crosses and collected eggs daily. Eggs take approximately 40 degree-days above a developmental threshold of 12 °C to complete development [68]. By moving eggs between constant temperatures and manipulating development, we ensured neonate larvae were available for the initiation of each experiment detailed below.

#### 4.3.3. Experimental Protocol

We undertook three series of assessments of both strains: the parental TWB3 and the TKF strain, on artificial diet, cabbage and kale in January 2013, August 2015 and November 2016, effectively after 2, 21, and 29 rounds of selection, respectively. For each assessment, newly hatched larvae (0–12 h old) were introduced to 1–2 cabbage or kale leaves placed on moist filter paper in Petri dishes (9 cm diameter, maximum 10 larvae/dish) as well as 10 neonates onto artificial diet from TKF and TWB crosses (see Table 2 below for a summary of how many Petri dishes and larvae were set up). All experiments were conducted in a 29 °C environment cabinet, 65%RH, 12:12 L:D (Snijders Scientific, model EB2E, Snijders Labs, Tilburg, The Netherlands). 

In all experiments, survival and weight of larvae were assessed 3–4 days later; at that temperature, larvae would have been about 51–68 degree-days old (late II or early III instar). In 2013, surviving larvae were transferred to large plastic containers (20 × 30 × 10 cm) lined with moist paper towel and provided with whole stems of food plant material ad libitum or placed in individual cups with artificial diet. In 2015 and 2016, all larvae were individualized into rearing cups after 4 days in Petri dishes with their respective diets. A subset of larvae was weighed (Mettler XS105 Dual Range balance) before being put into cups to estimate growth rates. There was sufficient artificial diet in a cup for larvae to complete development. Larvae on kale and cabbage were moved onto fresh leaf material daily until they completed development. The time to pupation was recorded and pupae were weighed once cuticle had hardened (within 24 h). Pupae were checked twice daily and newly emerged adults sexed.

The number of successful paired crosses varied amongst experiments and each produced a variable number of viable eggs. Consequently, the disposition of eggs amongst rearing diet treatments was unbalanced (Table 4).

### 4.4. Behavioural Assays

#### 4.4.1. In 2016

Leaves from kale and cabbage were cut into doughnut-like or annulus shape (the diameter of the outer circle was 5 cm and the inner circle 2.4 cm) and imbedded in agar in the lid of Petri dishes. Twenty newly hatched larvae from each strain were put in the centre of the inner circle in each treatment. Petri dishes were put on the top of paper cups and the cups were stood in trays with shallow water to record escapes. 

After three hours, the larvae on the edge of a Petri dish were counted and removed to a new Petri dish with the same set up. After another three hours, the larvae on the edge of the second Petri dish to which they had been transferred were counted. The assay assessed the likelihood of leaving as the other larvae stayed and were feeding on the leaf material. There were 6 Petri dishes for each strain.

#### 4.4.2. In 2019

Neonate larvae from six TKF crosses and seven TWB crosses were each offered a choice of cabbage vs. artificial diet or kale vs. artificial diet. In most cases, multiple Petri dish were set up with 2 discs of each diet type embedded in agar and 20 neonates were placed in a small well in the centre. After 48 h, the location of all larvae was recorded: on each diet type or elsewhere (agar/dish).

### 4.5. Statistical Analysis

For all analysis in performance assays, we used a mixed effects GLM. For each response variable (early larval weight gain, time to pupation, pupal weight, and time as a pupa), we fitted a model with Year, Strain, Food and, where relevant, Gender as factors with family as a random variable nested within strain, and interaction between Food and Strain was included in the model. In all cases, data distributions were not Gaussian and the GLMR function in R 4.0.2 [69] was used. Larval weight was expressed as *ln*(weight)/age in days and there were sufficient data for all three years. For pupae-related data, there were too few survivors on plant material in the first year (see results) and we only analysed data for 2015 and 2016.

For the behaviour assay data collected in 2016, essentially a no-choice experiment, the response variable was the proportion of larvae that had stayed on the plant disc at 3 h. A two-way ANOVA was performed after logit transformation of the response variable using R 4.0.2. [69]. 

For the behavioural data collected in 2019, the response variable was the numbers of larvae found on diet or plant (cabbage or kale) or on the dish, essentially a choice experiment. A multinomial model was fitted to the data using R 4.0.2. [69].

## Figures and Tables

**Figure 1 plants-10-00954-f001:**
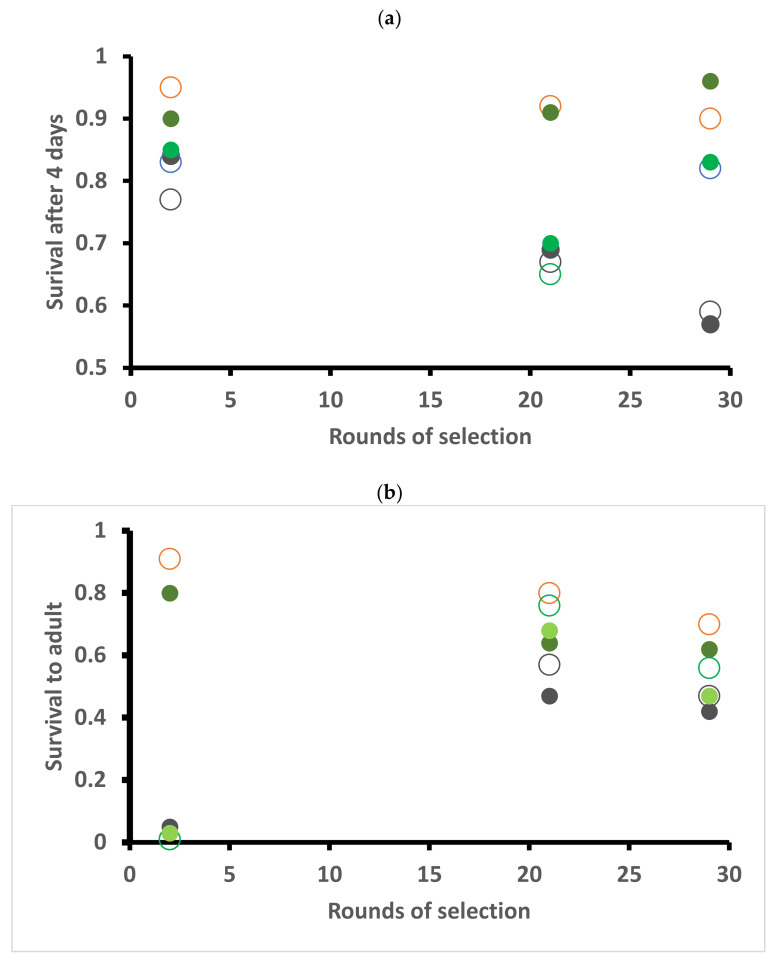
Percentage survival of early instars (**a**) and to the adult stage (**b**) of *Helicoverpa armigera* Toowoomba strain (large open circles) and Brassica-selected strain (solid smaller circles) when reared on artificial diet (brown), cabbage (dark green) or kale (light green), assessed at the indicated rounds of selection (Appendix A).

**Figure 2 plants-10-00954-f002:**
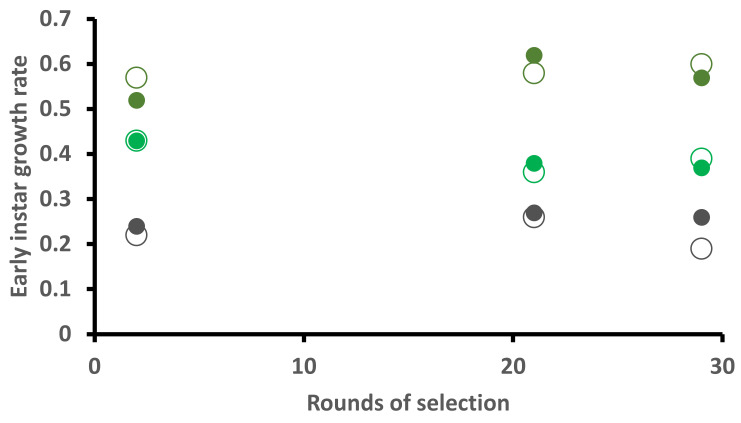
Growth rate (*ln* weight gain/time in days) of early instars of *Helicoverpa armigera* Toowoomba strain (large open circles) and Brassica-selected strain (solid smaller circles) when reared on artificial diet (brown), cabbage (dark green) or kale (light green) assessed at the indicated rounds of selection (Appendix A).

**Figure 3 plants-10-00954-f003:**
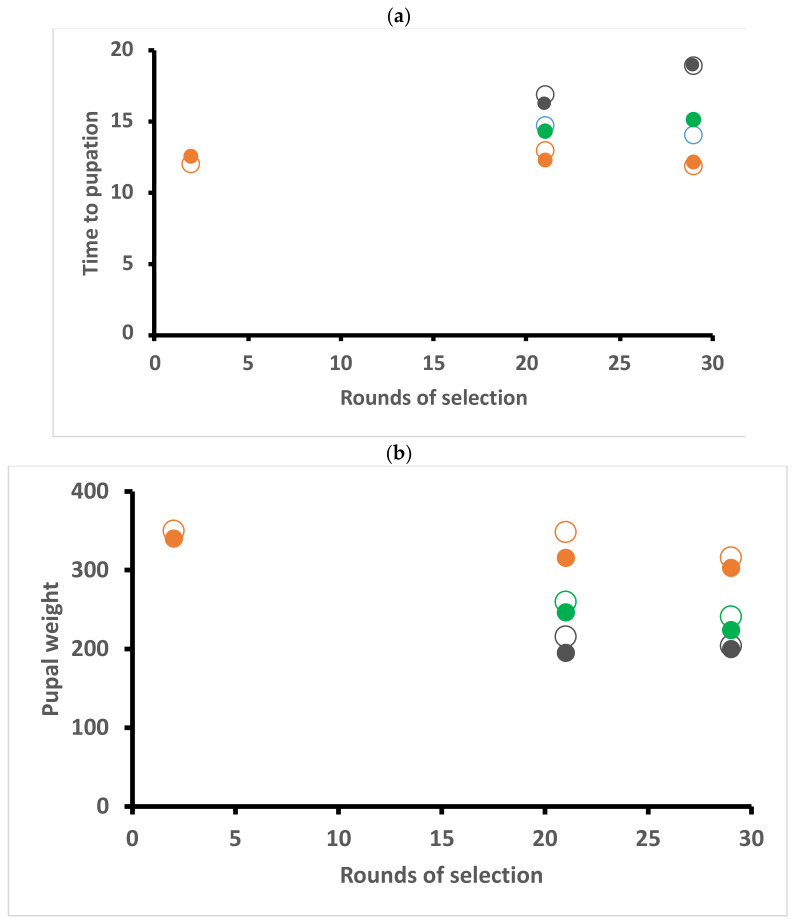
Time to pupation in days (**a**) and pupal weight in mg (**b**) of *Helicoverpa armigera* Toowoomba strain (large open circles) and Brassica-selected strain (solid smaller circles) when reared on artificial diet (brown), cabbage (dark green) or kale (light green) assessed at the indicated rounds of selection. Performance on artificial diet is plotted for the second selection round but were not used in the analysis (Appendix A).

**Table 1 plants-10-00954-t001:** Brassicaceae (Cruciferae) host plant use recorded by lepidopteran families with insect species classified as specialist, oligophagous or polyphagous. The number of pest species, mean number of host plant families (range is min and max), mean % families that are Brassicaceae (Br), mean number of host plant species and mean % of host species that are Brassicaceae (Br). See Appendix A for species details.

Family	Specialist	Oligophagous	Polyphagous	Total
Arctiidae		6	17	23
Cosmopterigidae			1	1
Gelechiidae	2	1	1	4
Geometridae	6	7	5	18
Hepialidae			3	3
Lasiocampidae		1		1
Lecithoceridae		1		1
Limacodidae			1	1
Lymantriidae	1		8	9
Lyonetiidae			1	1
Noctuidae	2	25	97	124
Nymphalidae	1	2	1	4
Papilionidae			1	1
Pieridae	34	15		49
Psychidae			2	2
Pyralidae	21	5	17	43
Sphingidae		2	2	4
Tineidae	1	1		2
Tortricidae	1		11	12
Yponomeutidae	14			14
Total	83	66	168	317
Pest Species	11	1	21	
Host Plant Families	1.8 (1–11) *	3.4 (2–8)	16.3 (3–69)	
% Families (Br)	78 (9–100)	35 (13–50)	9 (1–33)	
Host plant species	13 (1–115) *	8.2 (2–54)	57.6 (7–458)	
% species (Br)	93 (56–100)	27 (2–50)	6 (1–23)	

* the outlier is *Plutella xylostella* that is recorded on numerous other families and host species but we retained it as a “specialist” due to its characteristic behaviour towards GSLs and their hydrolysis products [26,27].

**Table 2 plants-10-00954-t002:** Percentage of first instar *Helicoverpa armigera* larvae from selected TKF and the parental TWB strain found on artificial diet, cabbage or kale, and other (agar or Petri dish lid) after 48 h.

Strain	Location Where Larvae Found:			
	Diet	Cabbage	Kale	Other
**TKF**	51%	26%		23%
	42%		41%	17%
**TWB**	48%	26%		26%
	44%		34%	22%

**Table 3 plants-10-00954-t003:** Performance measures (Larval development time in days, Pupal development time in days and Pupal weight in mg) of *Helicoverpa armigera* on a range of host plants under laboratory conditions at a range of temperatures (°C). For multiple studies, we report the range in mean values and just the mean if only one study could be found.

Host	Family	Temperature	Larval Time	Pupal Time	Pupal Weight	Study
Chrysanthemum	Asteraceae	25	20.7		315	5
Sunflower	Asteraceae	27	13.9		200–220	8, 9
Bean	Fabaceae	27	16.62	9.75	257	7
Chickpea	Fabaceae	25	15.6	14.6	260	2
Pigeonpea	Fabaceae	22.5–27	17.5–36.5	13.7–16.2	113–284	11, 12
Cajanus sp	Fabaceae	27	24.8	15.9	207	11
Cajanus sp	Fabaceae	27	35.9	16.8	170	11
Cowpea	Fabaceae	25			210–350	4
Soybean	Fabaceae	25	14.5		300	1, 5
Cotton	Malvaceae	25–27.5	11.4–22.8	10.1–14.2	244–337	1, 5, 6, 7
G. arboretum	Malvaceae	27.5	12.9–17.1	13–14	258–324	6
Okra	Malvaceae	25	13–14.8	14	248–340	1, 3, 5
Corn	Poaceae	25–27	14.5–16.5	9.6–14.2	160–350	3, 4, 7, 10
Chilli Pepper	Solanaceae	27	19.5–21.4	9.79	173–208	7
Eggplant	Solanaceae	25	19.85	14.01	270	3
Pepper	Solanaceae	25	14.1–21.2	14.3	267–290	3,5
Tobacco	Solanaceae	27	15–19.5	10	230–310	7, 8, 9
Tomato	Solanaceae	25–27	13.9–23	9–13.5	167–310	1, 3, 5, 7
Cabbage	Brassicaceae	28–30	17–25	8.6–11	200–224	This study, 13
Kale	Brassicaceae	28–30	14–15.2	8.2	224–241	This study

**Table 4 plants-10-00954-t004:** *Helicoverpa armigera* single pair matings (moth families) of the Toowoomba strain (TWB3) and cabbage selected strain (TKF) that produced enough fertile eggs to run assays for each assessment (moth family number is arbitrary) in 2013, 2015 and 2016, showing the initial number of first instars larvae exposed to each diet type (AD = artificial diet). Divide the number by 10 to obtain the number of Petri dishes.

2013							
Family Number	Larvae Exposed	Larvae Exposed	Larvae Exposed	Family Number	Larvae Exposed	Larvae Exposed	Larvae Exposed
TKF	AD	Cabbage	Kale	TWB	AD	Cabbage	Kale
2	20	40	40	12	20	45	40
3	10	10	20	15	10	10	10
4	25	80	85	17	20	60	60
5	10	40	40	19	7	50	40
9	10	20	20				
10	10	20	20				
**Total**	**85**	**210**	**225**		**57**	**165**	**150**
**2015**							
**Family Number**	**Larvae Exposed**	**Larvae Exposed**	**Larvae Exposed**	**Family Number**	**Larvae Exposed**	**Larvae Exposed**	**Larvae Exposed**
**TKF**	**AD**	**Cabbage**	**Kale**	**TWB**	**AD**	**Cabbage**	**Kale**
11	60	55	55	2	60	90	120
15			15	6	20	15	
19	60	16	60	17	20	7	10
24	100	100	100				
**Total**	**220**	**171**	**230**		**100**	**112**	**130**
**2016**							
**Family Number**	**Larvae Exposed**	**Larvae Exposed**	**Larvae Exposed**	**Family Number**	**Larvae Exposed**	**Larvae Exposed**	**Larvae Exposed**
**TKF**	**AD**	**Cabbage**	**Kale**	**TWB**	**AD**	**Cabbage**	**Kale**
427	20	30	30	504	20	30	30
429	20	30	30	507	20	30	30
432	20	30	30	510	20	30	25
439	20	30	30	513	20	30	30
445	20	30	30	524	20	30	30
				526	10	30	30
Total	100	150	150	Total	110	180	175

## Data Availability

Data reported in this study can be found in the Appendix A.

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
