# Peer review of "A Generalist Feeding on Brassicaceae: It Does Not Get Any Better with Selection"

_plants, 2021, doi:10.3390/plants10050954_

Round 1
Reviewer 1 Report
Dear Authors,
Paper is well presented, with interesting topic. If you just correct section order, it will be ok. So, start with 1 as introduction, 2 as material and methods, etc.
Lines 25-79 – Please use the same letter size
Line 62: change domesticated to cultivated
Line 80 - Can you also mention which part of the world is included in the survey or please be more accurate. Some species of Lepidoptera don't occure in Europe, and others are present only in Europe.
Line 274: be more accurate by expression : »see Fig… Which Figures?
Line 432: Please add reference for R 4.02.
Author Response
Materials and methods is after the Discussion as per Plants template
Lines 25 - 79 Font size corrected for Introduction
Line 62 Now line 64 changed to cultivated.
global survey included line 85
Figures 1-3 added to line 280
Reference for R added
Reviewer 2 Report
This is overall a very relevant and interesting paper. Although the result is negative, it should definitely be published. In fact, thorough negative results should also be published of course. In this case, the negative result is accompanied by an interesting survey of host plant specialization and essay-like discussions of what constitutes a pest etc.
I have the following criticism/comments:'
Title, abstract and first lines, line 49, line 62, etc. The genus Brassica and the family Brassicaceae seems to get mixed up a bit. This is commonly done and makes a lot of confusion, especially since it is well-etsblished that the genus Brassica is not monophyletic and probably ought to be revised, split and pooled with other genera, if only it wasn't such a well-established name. But I think the authors should be a bit more distinct about what taxon is discussed. And perhaps provide a bit of an introduction to the phylogeny of the genus Brassica, if you wish to still have a discussion mentioning this "type genus" of the family.
line 33, glucosinolate products. In a text about the biochemical mechanism of myrosinase, it would make sense to state that ITCs and the other three mentioned functionalities result from GSL hydrolysis. But I see this text more centered on what an insect will face when chewing mustard plants, and in that sense I think the main final products should be listed. Hence, I think that ascorbigens, other indoles and oxazolidine-2-thiones should be mentioned here. Am I wrong if I guess that close to 100% of Brassicas will form most of these, and many would form all of them?
line 37-38: there is a change of font-size here.
line 63. In order to make this paper readable on its own, could you give just one sentence of approximately how the two GSL profiles differ? (kale versus cabbage)
line 64-66. Here or somewhere else (results, experimental, I would have liked a bit more of a description of how the selection process occurred. Is the selection leading to the Kohlfütterung line what is described in section 4.3.2? Could you make thsi even more clear in the text? e.g. refer to section 4.3.2. from line 67?
table 1.
I am not sure why you would count how many of the host plant species that are from Brassicaceae, as the taxon characterized by GSLs is rather the Brassicales. And I am not even sure how the line "% families (Br) 78 (9-100) 35 (13-50) 9 (1-9)" was calculated. How can 78% of a number of families be just one family. Did you calculate % families that are Brassicales?
line 98. I could not find the supplementary data referred to.
line 100. Perhaps provide an argument here, such as the experimentally verified attraction to GSLs and ITCs?
Line 310 here, definately it is unclear whether you mean the genus Brassica or the family Brassicaceea (or the order Brassicales). In general, I think we should avoid writing brassica without capitals, whatever the intended meaning. The additional use of the synonym Cruciferae adds to the confusion, not all readers will be trained plant taxonomists... :-)
line 363 9 cm with space
line 366 move degree symbol a bit.
line 337 space between number and degree-symbol.
Author Response
Title, abstract and first lines, line 49, line 62, etc
In line with these comments we have changed the title to Brassicaceae rather than Brassica as well as in the first line of the introduction. We take the point about the phylogeny of the Brassicales, but that is beyond the scope of this paper. We also feel that the use of the common term, brassicas, is appropriate as cabbage and kale are in that genus.
line 33, glucosinolate products. The following text has been inserted after line 30
"Although GSL are themselves not toxic, when brought together with activating myrosinases that are maintained in separate compartments in plant cells, for example by the actions of a chewing herbivore, they are hydrolysed to form an array of products. These include more prominently the toxic isothiocyanates (ITCs), the main components of the so-called mustard oil bomb [4], as well as other hydrolysis products with potential biological activity, namely epithionitriles, simple nitriles and organic thiocyanates, depending on the reaction cofactors and the GSL side chain [5-7]. Due to the reactivities of some of these molecules, especially of ITCs, further intramolecular rearrangements and conjugations to biological nucleophiles are common, leading for example to cyclization derivatives, as well as indole carbinols and ascorbigens [5-7]. The release of these toxic compounds upon plant tissue....."
line 37-38: there is a change of font-size here. Changed.
line 63...following was added (now line 85)
Cabbage with 3-methylsulfinylpropyl (3MSOP)- GSL (absent from kale) and in kale 2-hydroxy-3-butenyl- (2OH3But) (absent from cabbage).
line 64-66.
e.g. refer to 4.3.2 added see line 90
I am not sure why you would count how many of the host plant species that are from....
The percentages are lepidopteran families so the data is correct.
line 98. I could not find the supplementary data referred to.
Supplementary data is now uploaded.
line 100. Added the following. Now line 123
due to its characteristic behaviour towards GSLs and their hydrolysis products [26,27].
Line 310 here,
All brassica changed to Brassica.
We have added the following line and reference (341). regarding the use of Cruciferae.
It should be noted that this data base uses the old classification, Cruciferae, which is now Brassicaceae [66].
line 363 9 cm with space. Done
line 366 move degree symbol a bit. Done
line 337 space between number and degree-symbol. Done